# Analysis of Volatile Compounds in Different Varieties of Plum Fruits Based on Headspace Solid-Phase Microextraction-Gas Chromatography-Mass Spectrometry Technique

Qin Zhang [1], Shouliang Zhu [2], Xin Lin [1], Junsen Peng [1], Dengcan Luo [1], Xuan Wan [1], Yun Zhang [1], Xiaoqing Dong [1,*] and Yuhua Ma [3,*]

1   Fruit Crops Center of Guizhou Engineering Research, College of Agricultural, Guizhou University, Guiyang 550025, China; zhangqin982022@163.com (Q.Z.); yjt.joe@foxmail.com (D.L.)
2   Guizhou Workstation for Fruit and Vegetables, Guiyang 550025, China
3   Guizhou Academy of Agricultural Sciences, Guiyang 550006, China
*   Correspondence: xqdong@gzu.edu.cn (X.D.); myh79@163.com (Y.M.)

**Abstract:** To investigate the differences in the volatile compounds of plum fruit samples from different cultivars, the volatile compounds of the 'Fengtang' plum, 'Kongxin' plum, and 'Cuihong' plum fruits were analyzed using headspace solid-phase microextraction–gas chromatography–mass spectrometry (HS-SPME-GC-MS). The results demonstrated that a total of 938 volatile compounds were identified in three plum fruits, including 200 terpenoids, 171 esters, 116 heterocyclic compounds, 89 hydrocarbons, 82 ketones and alcohols, 63 aldehydes, 54 aromatic hydrocarbons, 21 amines, 18 acids, 17 phenols, 10 nitrogenous compounds, 7 sulfur compounds, and other compounds, 470 of which were common to all the cultivars. Moreover, 704, 691, and 704 volatile substances were detected, respectively, in the 'Fengtang' plum, 'Kongxin' plum, and 'Cuihong' plum, with terpenoids, esters, and heterocycles as the main compounds, accounting for 62.12~72.03% of the volatile compounds. The results of principal component analysis (PCA) and cluster analysis (CA) illustrated that the 'Fengtang' plum and 'Cuihong' plum were similar in terms of volatile compounds; the 'Kongxin' plum compounds were different from those in the other cultivars. Orthogonal partial least squares discriminant analysis was performed, revealing the typical volatile compounds that differed among the plum fruits of the different varieties; thus, the three plum fruits could be better distinguished. These results can provide a theoretical basis for the studies of plum fruit flavor, quality, and geographical origin identification.

**Keywords:** plum fruit; different cultivars; volatile compounds; HS-SPME-GC-MS; variance analysis

## 1. Introduction

The plum (*Prunus salicina* L.) is a perennial deciduous fruit tree of the genus *Prunus* in the family Rosaceae; it is native to China, with a cultivation history of more than 3000 years, and is the second largest drupe fruit tree after the peach [1]. Plum fruits are colorful, sweet, sour, juicy, and high in nutritional value, and they are loved by consumers because they contain sugar, organic acids, amino acids, vitamins, fats, proteins, and minerals, as well as anthocyanins, flavonoids, phenolic compounds, and other antioxidant substances [2]; these substances have high antioxidant activity [3] and can promote stomach and intestinal digestion. In recent years, with the continuous improvement of living standards, people's requirements for the quality and flavor of plum fruits have also increased, and volatility is an important factor affecting the flavor of plum fruits.

Volatile compounds are important components of fruit flavor quality and directly affect consumer preferences [4]. The volatile compounds of fruits not only reflect the characteristics of different fruit but also directly affect the sensory quality of fresh fruits and their processed products [5], and they are important indicators for fruit quality evaluation [6]. A volatile substance is composed of a complex mixture of multiple volatile

compounds that are constantly changing during plant growth and are further formed into volatile compounds from a large number of nonvolatile plant precursors through various biochemical pathways [7,8]. It has been shown that the volatile compounds of plum fruits are mainly composed of esters, alcohols, aldehydes, ketones, acids, terpenoids, and other compounds [9], and the types and contents of volatile substances vary greatly among the different varieties [10]. Lozano et al. (2009) showed that guanidine, 3-hexen-1-ol, 4-hexen-1-ol acetate, and hexyl acetate were the major volatile components of six plums [11]. Li et al. (2019) found that the volatiles of the fruit of Cerasus humilis (Bge.) sok were composed of eight classes of compounds, such as esters, alkanes, alcohols, aromatics, and aldehydes, with esters being the main volatile compounds [12]. On the other hand, Wang et al. (2012) showed that the volatile substances of the 'Black Gem' plum were alcohols, aldehydes, esters, ketones, acids, phenols, and nine other classes of compounds, among which (Z)-3-hexen-1-ol, 1-hexanol, benzaldehyde, γ-dodecalactone, and hexadecanoic acid were important for the volatiles of the fruit of the 'Black Gem' plum, and that the volatile formation had an important effect [13]. Yu et al. (2012) found that the most abundant volatile component among the six plum fruits was 'Aus plum 13', in which the relative contents of the volatile compounds were hexane, 2-nonenol, 2-butoxyethanol, and hexanal [14]. At present, studies on the volatile composition and content of the fruits of the 'Fengtang' plum, 'Kongxin' plum, and 'Cuihong' plum have not been reported.

Thus, in this research, headspace solid-phase microextraction (HS-SPME) combined with gas chromatography–mass spectrometry (GC-MS) was used for the comparative analysis of the volatile substances in the fruits of the 'Fengtang' plum, 'Kongxin' plum, and 'Cuihong' plum. Principal component analysis (PCA) and cluster analysis (CA) were used to compare the differences in the volatile compositions among the fruits of the three varieties of plum. In addition, orthogonal partial least squares discriminant analysis (OPLS-DA) and variable importance projection (VIP) were used to determine the major differences in the volatile components among the fruits of different varieties of plums in order to provide scientific evidence for the further study of plum fruits' flavors and qualities.

## 2. Materials and Methods

### 2.1. Materials and Reagents

As the research objects, the fruits of the *Prunus salicina* 'Fengtang' plum, *Prunus salicina* 'Kongxin' plum, and *Prunus salicina* 'Cuihong' plum (referred to as F, K, and C) were harvested from the well-managed orchards of each origin on 23 June 2021, 23 July 2021, and 20 August 2021, respectively. The 'Fengtang' plum was picked from the orchard of the Liuma 'Fengtang' plum Planting Farmers' Specialized Cooperative in Guizhou Province (105.52° E, 25.37° N); the 'Kongxin' plum was picked from the orchard of the Farmers' Specialized Cooperative in Yanhe Tujia Autonomous County, Guizhou Province (108.32° E, 28.32° N); the 'Cuihong' plum was harvested from the Baiyi Experimental Base, Institute of Fruit Tree Research, Guizhou Academy of Agricultural Sciences (106.59° E, 26.4° N). Immediately after the harvest, the fruits were transported back to the laboratory and placed at 4 °C and 85–90% relative humidity for 12 h to dissipate the heat of the field; those without mechanical damage and with no pests or diseases were selected. All samples were promptly frozen in liquid nitrogen and stored in a −80 °C ultra-low temperature refrigerator for the HS-SPME-GC-MS analysis. Sodium chloride (analytical purity), Sinopharm Group Chemical Reagent Co., Ltd, Beijing, China; n-hexane (chromatographic purity), Merck, Darmstadt, Germany; standard (chromatographic purity), Sigma-Aldrich Company, Shanghai, China.

### 2.2. Instruments and Equipment

The following equipment was used: the 8890-7000D GC-MS coupled instrument, DB-5MS column (30 m × 0.25 mm × 0.25 μm), and 120 μm DVB/CWR/PDMS extraction head, Agilent, Shanghai, China.; an MM400 ball mill, Retsch GmbH, Haan, Germany the SPME Allow solid-phase microextraction device, Fiber SPME Allow solid phase microextraction

unit, fiber conditioning station, and agitator sample heating chamber, CTC Analytics AG, Zwingen, Switzerland.

*2.3. Test Method*

2.3.1. Sample Pretreatment

The fruit samples of the 'Fengtang' plum, 'Kongxin' plum, and 'Cuihong' plum were ground in liquid nitrogen and vortex-mixed well; 0.5 g of each sample was weighed. The samples were added to a 15 mL headspace vial containing 10 μL saturated sodium chloride solution, and the vial was sealed with a jaw cap equipped with a silicone headspace spacer. The headspace vials were shaken at 60 °C for 5 min and then subjected to fully automated headspace solid-phase microextraction HS-SPME sample extraction.

2.3.2. HS-SPME Conditions

The extraction head was aged in a fiber conditioning station at 250 °C for 5 min before sampling, and the 120 μm DVB/CWR/PDMS extraction head was inserted into the sample headspace vial; the headspace was extracted for 15 min and resolved at 250 °C for 5 min, followed by GC-MS separation and identification.

2.3.3. GC-MS Conditions

GC conditions: inlet temperature 250 °C; no split injection; solvent delay 3.5 min; high purity helium (purity not less than 99.999%) as a carrier gas; and constant flow rate 1.2 mL/min. Programmed ramp-up: start at 40 °C; hold for 3.5 min; ramp-up to 100 °C at 10 °C/min; ramp-up to 180 °C at 7 °C/min; and, finally, ramp-up to 280 °C at 25 °C/min and hold for 5 min. The temperature was increased to 280 °C at 25 °C/min and held for 5 min.

MS conditions: electron bombardment ion source (EI); ion source temperature 230 °C; quadrupole temperature 150 °C; mass spectrometry interface temperature 280 °C; electron energy 70 eV; scanning mode selected as ion detection mode (SIM); and qualitative and quantitative ion precision scanning.

*2.4. Statistical Analysis*

Based on the MWGC database, the metabolites of the samples were characterized by mass spectrometry, and the relative content of each compound component was determined by the internal standard method and the peak area normalization method for quantification [15]. The volatile substances of the three varieties of plum fruits were compared two by two and recorded as F vs. C, C vs. K, and F vs. K, respectively; orthogonal partial least squares discriminant analysis was performed by SMICA 14.1 and VIP values were calculated; analysis of variance (ANOVA) was performed by using SPSS 22.0 software, and the differences in volatile compounds were screened in each group with the conditions of significant difference at $p < 0.05$ and highly significant difference at $p < 0.01$ combined with VIP $\geq$ 1; principal component analysis and cluster analysis were performed by using R 3.6.3 software. For the compounds, principal component analysis and cluster analysis were performed using R 3.6.3 software.

**3. Results and Discussion**

*3.1. Analysis of Volatile Species of Plum Fruits of Different Varieties*

As shown in Figure 1, the HS-SPME-GC-MS conditions used could better separate the volatile compounds of the fruits of the three varieties of plums. As shown in Table 1, a total of 938 volatile substances in 14 categories were detected in the fruit samples of the three varieties of plums, including 200 terpenoids, 171 esters, 116 heterocyclics, 89 hydrocarbons, 82 ketones and alcohols, 63 aldehydes, 54 aromatic hydrocarbons, 21 amines, 18 acids, 17 phenols, 10 nitrogenous compounds, 7 sulfur-containing compounds, and 8 other categories. In the fruits of the different varieties of plums, 704 aromatic substances, including 154 terpenes, 120 esters, 71 hydrocarbons, 60 alcohols, 60 ketones, 51 aldehydes, 12 acids,

9 phenols, and 13 amines, were detected in both the 'Fengtang' plum and the 'Cuihong' plum. Phenols, 13 amines, 37 aromatic hydrocarbons, 101 heterocyclic compounds, 7 nitrogenous compounds, 3 sulfurous compounds, and 6 others were also detected. In the 'Kongxin' plum, 691 volatile substances were detected, including 155 terpenoids, 124 esters, 58 hydrocarbons, 61 alcohols, 55 ketones, 48 aldehydes, 13 acids, 13 phenols, 16 amines, 41 aromatic hydrocarbons, 92 heterocyclic compounds, 7 nitrogenous compounds, 4 sulfur-containing compounds, and 4 others. In addition, among the 938 volatile compounds, 470 volatile substances were common to the three plum fruits, accounting for 50.10% of the total volatile substances, with the esters and terpenoids dominating.

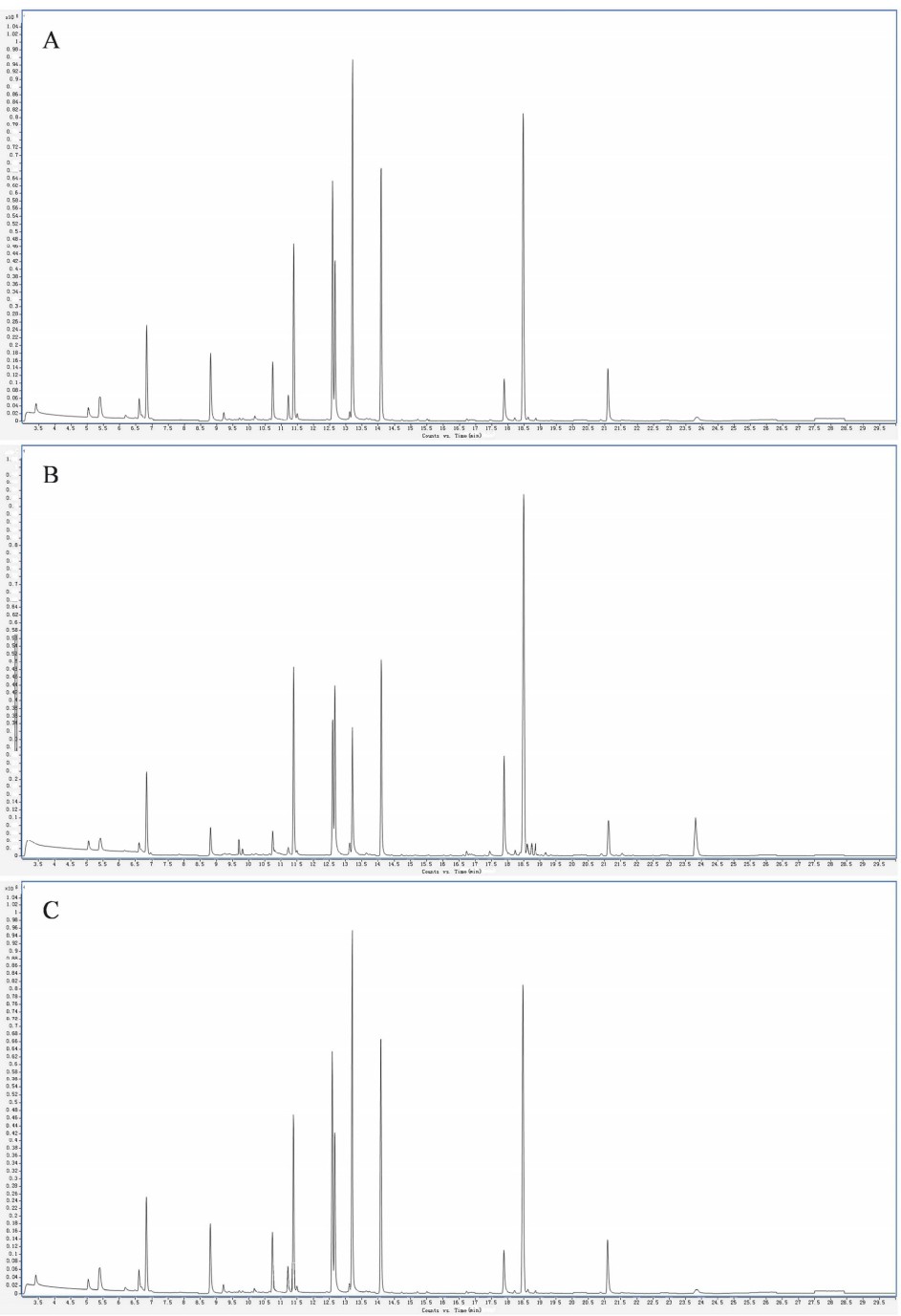

**Figure 1.** Total ion flow diagram for QC sample nature spectrum analysis. Note: (**A**, **B**, and **C**) are total ion flow diagrams for 'Fengtang' plum, 'Kongxin' plum, and 'Cuihong' plum, respectively.

**Table 1.** Types of volatile substances in plum fruits of different varieties.

| Compound Type | Total Volatile Compounds | Total Volatile Common Substances | Types and Amounts of Volatile Substances in Plum Fruits of Different Varieties | | |
|---|---|---|---|---|---|
| | | | 'Fengtang' Plum | 'Kongxin' Plum | 'Cuihong' Plum |
| Terpenoids | 200 | 109 | 154 | 155 | 154 |
| Esters | 171 | 77 | 120 | 124 | 120 |
| Hydrocarbons | 89 | 41 | 71 | 58 | 71 |
| Alcohols | 82 | 39 | 60 | 61 | 60 |
| Ketones | 82 | 35 | 60 | 55 | 60 |
| Aldehydes | 63 | 37 | 51 | 48 | 51 |
| Acids | 18 | 8 | 12 | 13 | 12 |
| Phenols | 17 | 5 | 9 | 13 | 9 |
| Amines | 21 | 8 | 13 | 16 | 13 |
| Aromatics | 54 | 24 | 37 | 41 | 37 |
| Heterocyclic | 116 | 78 | 101 | 92 | 101 |
| Nitrogen-containing compounds | 10 | 5 | 7 | 7 | 7 |
| Sulfur-containing compounds | 7 | 1 | 3 | 4 | 3 |
| Other categories | 8 | 3 | 6 | 4 | 6 |
| Total | 938 | 470 | 704 | 691 | 704 |

### 3.2. Analysis of Volatile Content of Plum Fruits of Different Varieties

A total of 938 volatile compounds were detected in the plum fruit samples of the three varieties, as shown in Table 2; terpenoids, esters, and heterocycles accounted for 65.80% to 68.71% of the total volatile compounds. It can be seen that these types of substances are the main volatile compounds that determine the volatile characteristics of plum fruits. Terpenoids accounted for 34.88% to 38.31% of the total volatile compounds and were mainly produced by the carbohydrate metabolic pathway [16]. Among them, the 'Kongxin' plum had the most terpene compounds, followed by the 'Cuihong' plum and 'Fengtang' plum, with little difference. The heterocyclic volatile compounds accounted for 14.67–20.78% of the total volatile compounds, but the content of heterocyclic compounds in the volatile compounds varied greatly among the varieties and the trend was opposite to that of the terpenoids, with the content of the 'Cuihong' plum = 'Fengtang' plum > 'Kongxin' plum. Ester compounds accounted for 12.57% to 12.94% of the total volatile compounds, in the following order: 'Cuihong' plum > 'Kongxin' plum > 'Fengtang' plum. Fruit aroma substances, such as esters, belong to the secondary metabolites of fruits, which are formed by fatty acids, amino acids, carbohydrates, etc., as precursor substances through the metabolism of amino acids, as well as the fatty acid metabolism pathway and a series of enzymatic reactions during the growth and development of fruits [17,18]. Alcohols accounted for 7.58% to 7.65% of the total volatile compounds, which were mainly hydrolyzed glycosides, amino acid, and fatty acid metabolites [19]; ketones accounted for 1.92% to 2.87% of the total volatile compounds, with the largest number of carotenoid degradation products originating from the fatty acid metabolism pathway [18]; ketones and alcohols, although they were both 82, each accounted for a large difference in the total amount of volatile compounds. In addition, the numbers of hydrocarbon, aldehyde, and aromatic compounds, although all more than 50, were not high in relative content, ranging from 4.82% to 5.54%, 5.48% to 6.95%, and 3.67% to 7.23%, respectively. The phenol and nitrogen compounds were both low, totaling 0.93% to 2.93%. The relative content of acids, amines, sulfur compounds, and other compounds in the plum fruits was extremely low, ranging from 0.01% to 0.67%. In summary, the volatile substances detected in the plum fruits of the three varieties contained 14 classes of substances, including terpenoids, esters, ketones, alcohols, aldehydes, hydrocarbons, phenols, and acids, among which terpenoids were the most dominant component, followed by heterocyclic compounds and esters again. The results of previous studies showed that the main volatile compounds in plum fruits

were aldehydes, esters, and ketones [20,21]; these findings may be related to the varieties of plum fruits and the different detection methods.

**Table 2.** Relative contents of various classes of volatile compounds in plum fruits.

| Compound Type | Relative Content % | | |
| --- | --- | --- | --- |
| | 'Fengtang' Plum | 'Kongxin' Plum | 'Cuihong' Plum |
| Terpenoids | 34.889 ± 0.077 b | 38.312 ± 0.148 a | 34.985 ± 0.052 b |
| Ester | 12.573 ± 0.078 b | 12.816 ± 0.150 ab | 12.946 ± 0.023 a |
| Hydrocarbons | 5.189 ± 0.073 b | 5.540 ± 0.033 a | 4.828 ± 0.060 c |
| Alcohol | 7.654 ± 0.099 a | 7.582 ± 0.071 a | 7.639 ± 0.044 a |
| Ketone | 2.089 ± 0.022 b | 2.872 ± 0.001 a | 1.922 ± 0.060 c |
| Aldehyde | 6.952 ± 0.069 a | 5.488 ± 0.057 b | 6.890 ± 0.035 c |
| Acid | 0.543 ± 0.011 b | 0.623 ± 0.017 a | 0.555 ± 0.007 b |
| Phenol | 2.930 ± 0.044 ab | 2.980 ± 0.040 a | 2.812 ± 0.010 b |
| Amine | 0.498 ± 0.038 a | 0.249 ± 0.004 b | 0.673 ± 0.034 c |
| Aromatics | 3.801 ± 0.012 b | 7.233 ± 0.013 a | 3.672 ± 0.022 c |
| Heterocyclic | 20.647 ± 0.061 a | 14.676 ± 0.076 b | 20.787 ± 0.023 a |
| Nitrogen compounds | 1.566 ± 0.006 b | 0.935 ± 0.009 c | 1.607 ± 0.007 a |
| Sulfur compounds | 0.636 ± 0.013 a | 0.663 ± 0.017 a | 0.654 ± 0.003 a |
| Other categories | 0.014 ± 0.0002 a | 0.007 ± 0.0003 c | 0.011 ± 0.0004 b |

Note: Data are shown as the mean ± S.E. Different letters in the same row differ significantly ($p < 0.05$).

In terms of individual volatile compounds, there were 34 compounds with high relative content >1.00%, as shown in Table 3, including 13 terpenoids, 6 heterocyclics, 4 aromatic hydrocarbons, 2 alcohols, 2 hydrocarbons, 3 aldehydes, 3 esters, and 1 phenol. Of these 34 volatile substances, 19 constituents were simultaneously present in the three plum fruits, namely (3-Bromo-1-methylpropoxymethyl)benzene, (5-bromopentyl)-Benzene, 1,3-Cyclohexadiene, 5-(1,5-dimethyl-4-hexenyl)-2-methyl-, [S-(R*,S*)]-, cis-.alpha.-Bisabolene, Eremophilene, (1S,2E,6E,10R)-3,7,11,11-Tetramethylbicyclo[8.1.0]undeca-2,6-diene, .alpha.-Muurolene, Bicyclo[2.2.1]heptan-2-ol, 1,7,7-trimethyl-, (1S-endo)-, Carvenone, Aristolochene, 2-Hexanoylfuran, Furo[3,4-c]pyridin-1(3H)-one, 7-hydroxy-6-methyl-, 1H-Pyrazole-1-carboximidamide, 3,5-dimethyl-, Ethanone, 1-(1H-pyrazol-4-yl)-, Cyclohexanemethanol, .alpha.,.alpha.-dimethyl-4-methylene-, 2-Hexenal, (E)-, 1,5-Cycloundecadiene, 8,8-dimethyl-9-methylene-, 1,3-Benzenediol, 4,5-dimethyl-, and ethyl acetate. Studies have shown that these compounds mainly exhibit natural green, floral, fruity, camphor, and pine aromas [21–24].

**Table 3.** Relative percentages of main volatile compounds in different varieties of plum.

| NO. | Compound | CAS | RT/Min | Relative Contents % | | |
| --- | --- | --- | --- | --- | --- | --- |
| | | | | 'Fengtang' Plum | 'Kongxin' Plum | 'Cuihong' Plum |
| | Terpenoids (13) | | | | | |
| 1 | 1,3-Cyclohexadiene, 5-(1,5-dimethyl-4-hexenyl)-2-methyl-, [S-(R*,S*)]- | 495-60-3 | 18.43 | 1.260 ± 0.013 b | 1.678 ± 0.008 a | 1.260 ± 0.005 b |
| 2 | cis-.alpha.-Bisabolene | 29837-07-8 | 18.44 | 1.188 ± 0.010 b | 2.053 ± 0.023 a | 1.189 ± 0.005 b |
| 3 | [1α,4aα,8aα]-1,2,4a,5,6,8a-hexahydro-4-7-dimethyl-1-[1-methylethyl]naphthalene | 31983-22-9 | 17.46 | LOD | 1.556 ± 0.019 | LOD |
| 4 | Eremophilene | 10219-75-7 | 18.42 | 1.195 ± 0.009 b | 1.525 ± 0.015 a | 1.183 ± 0.003 c |

Table 3. *Cont.*

| NO. | Compound | CAS | RT/Min | Relative Contents % | | |
|---|---|---|---|---|---|---|
| | | | | 'Fengtang' Plum | 'Kongxin' Plum | 'Cuihong' Plum |
| 5 | Bicyclo[2.2.1]heptane, 2-methyl-3-methylene-2-(4-methyl-3-pentenyl)-, (1S-endo)- | 25532-78-9 | 17.87 | LOD | 1.122 ± 0.040 | LOD |
| 6 | (1S,2E,6E,10R)-3,7,11,11-Tetramethylbicyclo[8.1.0]undeca-2,6-diene | 24703-35-3 | 18.35 | 1.061 ± 0.010 b | 1.390 ± 0.016 a | 1.063 ± 0.002 b |
| 7 | .alpha.-Muurolene | 10208-80-7 | 18.47 | 1.189 ± 0.010 b | 1.556 ± 0.019 a | 1.260 ± 0.005 c |
| 8 | Bicyclo[2.2.1]heptan-2-ol, 1,7,7-trimethyl-, (1S-endo)- | 464-45-9 | 12.58 | 2.027 ± 0.016 a | 1.231 ± 0.015 b | 2.057 ± 0.002 a |
| 9 | Isoborneol | 124-76-5 | 12.57 | LOD | 1.33 ± 0.015 | LOD |
| 10 | endo-Borneol | 507-70-0 | 12.53 | 2.030 ± 0.013 a | 1.231 ± 0.015 b | 2.063 ± 0.004 a |
| 11 | Carvenone | 499-74-1 | 14.09 | 3.105 ± 0.014 b | 0.960 ± 0.014 c | 1.767 ± 0.020 a |
| 12 | 5,7-Octadien-4-one, 2,6-dimethyl-, (Z)- | 3588-18-9 | 12.59 | 2.031 ± 0.014 a | LOD | 2.144 ± 0.017 b |
| 13 | (-)-Aristolene | 6831-16-9 | 17.88 | 1.189 ± 0.010 a | 1.228 ± 0.013 b | 1.191 ± 0.005 a |
| | Heterocyclic (6) | | | | | |
| 14 | 2-((3,3-Dimethyloxiran-2-yl)methyl)-3-methylfuran | 92356-06-4 | 12.61 | 2.009 ± 0.019 a | LOD | 2.039 ± 0.001 a |
| 15 | 2-Hexanoylfuran | 14360-50-0 | 14.08 | 2.849 ± 0.012 b | 2.455 ± 0.015 c | 2.985 ± 0.016 a |
| 16 | Furo[3,4-c]pyridin-1(3H)-one, 7-hydroxy-6-methyl- | 4753-19-9 | 13.48 | 1.019 ± 0.014 b | 1.310 ± 0.016 a | 1.017 ± 0.006 b |
| 17 | 1H-Pyrazole-1-carboximidamide, 3,5-dimethyl- | 22906-75-8 | 11.38 | 1.353 ± 0.041 b | 1.821 ± 0.031 a | 0.913 ± 0.001 c |
| 18 | Furaneol | 3658-77-3 | 13.21 | 1.296 ± 0.027 b | LOD | 1.363 ± 0.018 a |
| 19 | Ethanone, 1-(1H-pyrazol-4-yl)- | 25016-16-4 | 11.58 | 2.017 ± 0.009 a | 1.233 ± 0.015 b | 2.150 ± 0.013 c |
| | Aromatics (4) | | | | | |
| 20 | (3-Bromo-1-methylpropoxymethyl)benzene | 51666-29-6 | 18.46 | 1.678 ± 0.013 b | 2.226 ± 0.023 a | 1.679 ± 0.009 b |
| 21 | (5-bromopentyl)-Benzene | 14469-83-1 | 18.45 | 1.678 ± 0.013 b | 2.221 ± 0.023 a | 1.679 ± 0.008 c |
| 22 | Benzene, 1,2-dimethoxy-4-(1-propenyl)- | 93-16-3 | 18.34 | LOD | 1.098 ± 0.006 | LOD |
| 23 | Benzoic acid, 4-hydroxy- | 99-96-7 | 18.29 | LOD | 1.388 ± 0.016 | LOD |
| | Alcohol (2) | | | | | |
| 24 | Cyclohexanemethanol, .alpha.,.alpha.-dimethyl-4-methylene- | 7299-42-5 | 12.66 | 1.113 ± 0.009 | 1.120 ± 0.006 | 1.125 ± 0.005 |
| 25 | Benzenemethanol, .alpha.-2-propenyl- | 936-58-3 | 14.10 | 1.282 ± 0.004 b | LOD | 1.322 ± 0.006 a |
| | Aldehyde (3) | | | | | |
| 26 | 2,6-Nonadienal, (E,Z)- | 557-48-2 | 13.02 | LOD | 1.621 ± 0.014 | LOD |
| 27 | 2-Hexenal, (E)- | 6728-26-3 | 12.66 | 1.640 ± 0.006 b | 1.033 ± 0.006 c | 2.047 ± 0.003 a |
| 28 | 5-Heptenal, 2,6-dimethyl- | 106-72-9 | 10.73 | 2.304 ± 0.048 b | LOD | 2.429 ± 0.035 a |
| | Ester (3) | | | | | |
| 29 | Ethyl acetate | 141-78-6 | 19.01 | 2.136 ± 0.006 c | 2.028 ± 0.005 a | 2.043 ± 0.004 b |
| 30 | Benzyl tiglate | 37526-88-8 | 18.51 | LOD | 2.228 ± 0.023 | LOD |
| 31 | Isobornyl formate | 1200-67-5 | 14.12 | 1.998 ± 0.007 b | LOD | 2.170 ± 0.010 a |
| | Hydrocarbons (2) | | | | | |
| 32 | 1,5-Cycloundecadiene, 8,8-dimethyl-9-methylene- | 62338-54-9 | 18.49 | 1.638 ± 0.014 b | 2.058 ± 0.022 a | 1.636 ± 0.009 b |
| 33 | Albene | 38451-64-8 | 11.60 | LOD | 1.233 ± 0.015 | LOD |
| | Phenol (1) | | | | | |
| 34 | 1,3-Benzenediol, 4,5-dimethyl- | 527-55-9 | 18.48 | 1.578 ± 0.023 b | 2.757 ± 0.039 a | 1.064 ± 0.016 c |

Note: Data are shown as the mean ± S.E. Different letters in the same row differ significantly ($p < 0.05$). LOD: indicates not detected; same below. CAS: substance CAS number, Agilent database number. RT: retention time of compounds on nonpolar columns.

In terms of characteristic volatile compounds, the characteristic 'Kongxin' plum volatile compounds include benzene, 1,2-dimethoxy-4-(1-propenyl)-, benzoic acid, 4-hydroxy-, [1α,4aα,8aα]-1,2,4a,5,6,8a-hexahydro-4-7-dimethyl-1-[1-methylethyl]naphthalene, Bicyclo[2.2.1]heptane, 2-methyl-3-methylene-2-(4-methyl-3-pentenyl)-, (1S-endo)-, Isoborneol, Albene, 2,6-Nonadienal, (E,Z)-, and Benzyl tiglate. The eight compounds with high relative amounts of the substances were, respectively, Benzyl tiglate (2.22%), 2,6-Nonadienal, (E,Z)-(1.62%), and [1α,4aα,8aα]-1,2,4a,5,6,8a-hexahydro-4-7-dimethyl-1-[1-methylethyl]naphthalene(1.55%), of which 2,6-Nonadienal, (E,Z)- has a typical cucumber flavor and has been detected in a variety of fruits, such as kiwifruit, persimmons, and melons [25–27]. These characteristic volatile compounds may have important influences on the development of flavor differences among the plum fruit varieties. In addition, some of the volatile compounds, although detected in the fruits of different plum varieties, varied greatly in their contents and may also contribute to the formation of characteristic flavors among the varieties, with the most typical compounds being 1,3-Benzenediol and 4,5-dimethyl- in the fruits of the three varieties of plums, with relative contents of 1.57%, 2.75%, and 1.06%, respectively; the relative content of Carvenone was 3.10% in the 'Fengtang' plum and 1.76% in the 'Cuihong' plum. It was only 0.96% in the 'Kongxin' plum; also included were substances such as 2-Hexenal, (E)-, 1H-Pyrazole-1-carboximidamide, and 3,5-dimethyl-.

### 3.3. Principal Component Analysis of Volatile Substances of Plum Fruits of Different Varieties

PCA analysis is a statistical method to examine the correlation between multiple variables, retaining the original data as much as possible to reflect the information of the data more intuitively and simply [28]. As can be seen from Figure 2, the contribution of PC1 was 79.42% and the contribution of PC2 was 7.15%, with a cumulative contribution of 86.57%; thus, the overall information is reflected well. The groups of plum fruit samples were clustered together internally, with a clear trend of separation between different varieties, which generally reflected the volatile differences between different varieties of plum fruits and the PCA results. In addition, with the 'Fengtang' plum and 'Cuihong' plum on one side and the 'Kongxin' plum on the other side, they were basically effectively differentiated, indicating that the volatile compounds of the fruits of the 'Fengtang' plum and 'Cuihong' plum differed less in terms of type and relative content, whereas the differences were greater with the 'Kongxin' plum, suggesting that they had significantly different flavors [29].

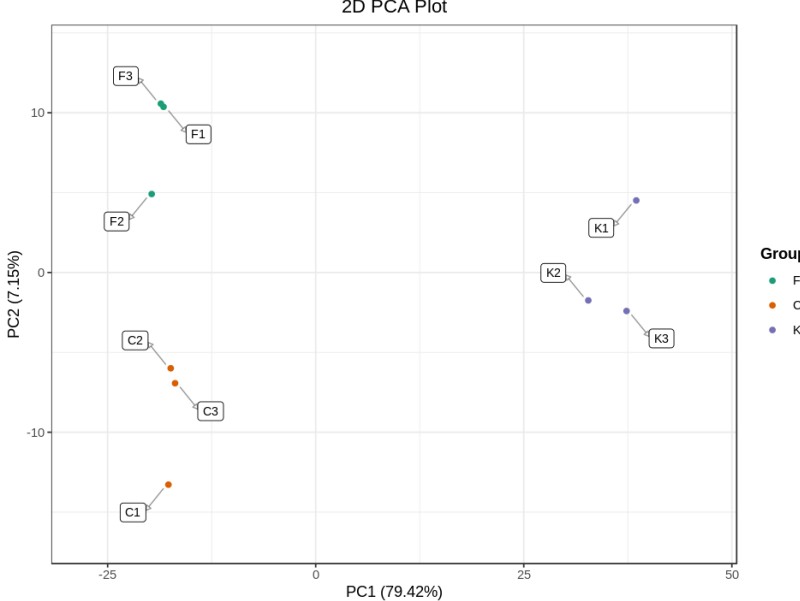

**Figure 2.** The PCA analysis diagram of different varieties of plum.

### 3.4. Cluster Analysis of Volatile Substances of Plum Fruits of Different Varieties

Cluster analysis is a method for describing differences between varieties; a certain degree of homogeneity in plum fruits with similar volatile characteristics is reflected, while different clusters reflect different similarities [30]. The clustering analysis enabled the substances with high similarity in different samples to be clustered together and the differences between samples to be visualized [31]. From the clustering results, Figure 3 shows that the clustering dendrogram of the volatile compounds of the different plum varieties shows two main categories, in which the samples of the 'Fengtang' plum and 'Cuihong' plum were grouped; the samples of the 'Kongxin' plum were in the other category, which indicated that the volatile compounds of the 'Fengtang' plum and 'Cuihong' plum were highly similar, while they were different from those of the 'Kongxin' plum; this corresponds to the PCA results, which showed differences in volatile composition among the fruit samples of the different varieties [32].

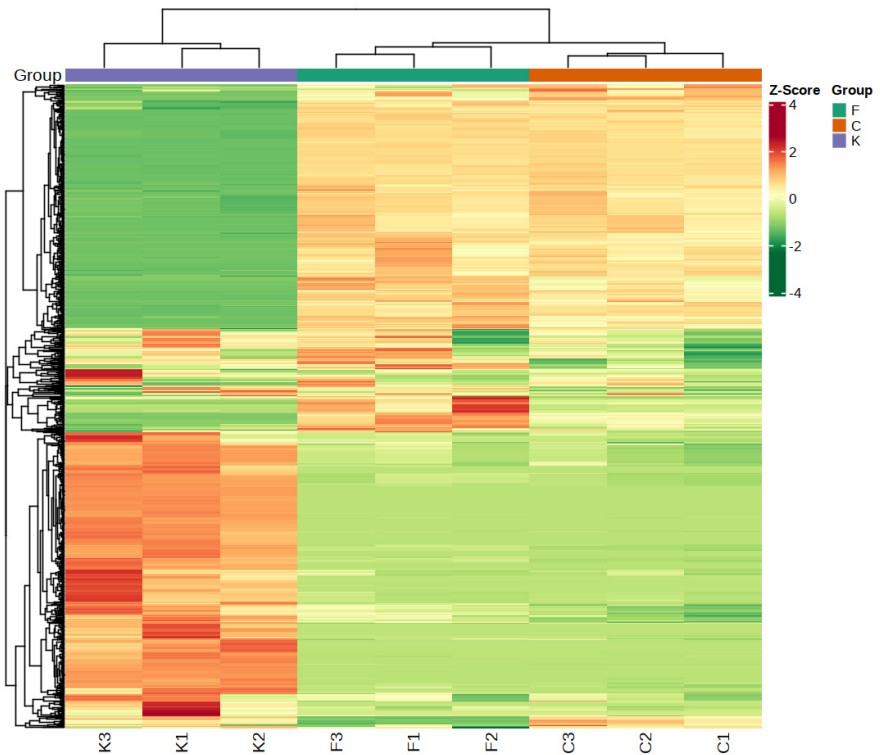

**Figure 3.** The cluster analysis of different varieties of plum.

### 3.5. Orthogonal Partial Least Squares Discriminant Analysis of Volatile Substances of Plum Fruits of Different Varieties

OPLS-DA belongs to supervised analysis, which can remove the influence caused by uncontrolled variables on the data as much as possible by presetting the classification, and can further explore the data information; at the same time, it can quantify the degree of differences between samples caused by the characteristic flavor substances and thereby reduce the error of the results [33]. To further confirm the differences between the fruit samples of different plum varieties, the volatile substances of the 'Fengtang' plum, 'Kongxin' plum, and 'Cuihong' plum fruits were subjected to OPLS-DA and two discriminant analyses, denoted as F vs. C, C vs. K, and F vs. K. From the OPLS score plots in Figure 4A,C,E, it can be seen that the contribution of PC1 to F vs. C was 34.20%, and the contribution of PC2 was 29.0%; the contribution of PC1 to F vs. K was 86.60%, and the contribution of PC2 was 5.77%; the contribution of PC1 to C vs. K was 87.20%, and PC2 had a contribution rate of 5.08%. All the groups showed an obvious separation trend, indicating that there are some differences in the volatile substances of the different varieties of plum fruits. From Figure 4B,D,F and Table 4, it can be seen that the indicators in the

evaluation parameters of the OPLS-DA model for each group were greater than 0.5 and $Q^2 > 0.9$. After 200 replacement tests, the results showed that the original $R^2$ and $Q^2$ of the three groups of models were greater than the corresponding values after the Y replacement, which indicated that there was no overfitting in the OPLS-DA discriminant model and that the model validation was effective and could be used in the subsequent characteristic determination of volatile substances.

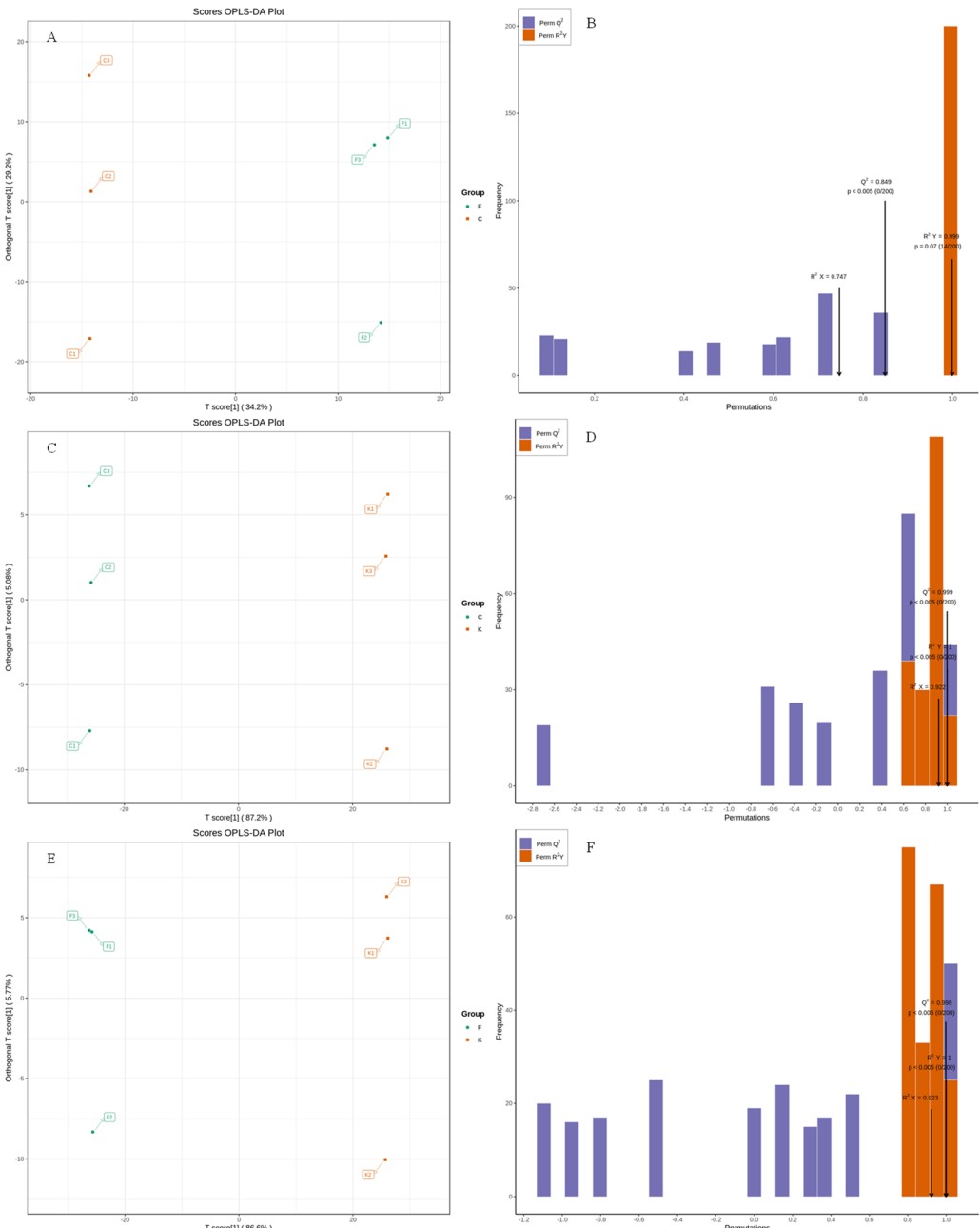

**Figure 4.** Score plot and validation of OPLS-DA model of volatile compounds in different plum cultivars. Note: (**A**,**B**; **C**,**D**; **E**,**F**) were OPLS-DA score plots and validation plots for F vsC, C vsK, and F vs K, respectively.

**Table 4.** Parameters of OPLS-DA model.

| Groups | $R^2X$ | $R^2Y$ | $Q^2$ |
|---|---|---|---|
| F vs. C | 0.747 | 0.999 | 0.849 |
| C vs. K | 0.922 | 1 | 0.999 |
| F vs. K | 0.923 | 1 | 0.998 |

The size of the VIP value of the key variables formed by OPLS-DA was used to analyze the key variables of the volatile compounds of the different varieties of plum fruits; the variables with VIP > 1 were considered to be the metabolites leading to the differences between groups [34]. According to the criteria of VIP > 1 and $p < 0.05$, the differential volatile substances of different varieties of plum fruits were screened, as shown in Table 5 below, and 23, 121, and 108 differential volatile substances were screened in F vs. C, C vs. K, and F vs. K, respectively, with the most differential compounds in C vs. K, the second highest in F vs. K, and the least differential substances in F vs. C. The substances with highly significant differences were indicated by $p < 0.01$ to further identify the characteristic volatile substances of the different varieties of plum fruits. The final screening results identified the six characteristic volatile substances in F vs. C as benzoic acid, methyl ester; 1,2-Benzenedimethanol; furan, 2-butyltetrahydro-; 1-Butanone, 2-hydroxy-1-phenyl-; acetic acid, phenoxy-; and tridecane, respectively. Among them, benzoic acid, methyl ester has a strong aromatic odor and can be used as a base fluid for perfumes [35]. A total of 46 characteristic volatiles were screened in C vs. K, including 8 esters, 12 terpenoids, 6 hydrocarbons, 4 alcohols, 5 ketones, 2 aldehydes, 1 phenol, 1 aromatic, and 7 heterocycles. The terpenoids predominantly contain humulene, iridomyrmecin, citronellol, 2,3-dehydro-1,8-cineole, .beta.-phellandrene, and .gamma.-terpinene, which has a rosy volatile [28]; humulene has a fruity and sweet volatile, and .gamma.-terpinene has a citrusy and sweet volatile [36]. In F vs. K, 40 characteristic volatiles were screened, including 10 esters, 5 terpenoids, 6 hydrocarbons, 2 alcohols, 4 ketones, 2 aldehydes, 2 acids, 2 aromatics, 6 heterocycles, and 1 nitrogen compound. The esters are predominantly esters containing substances such as benzoic acid, methyl ester, linalyl acetate, 2-butenoic acid, hexyl ester, benzeneacetic acid, 4-(1,1-dimethylethyl)-, methyl ester, acetic acid methyl ester, bornyl acetate, and ethyl acetate, which are floral and fruity [37], and global acetate has a cool pine volatile with a camphor-like scent [22].

**Table 5.** VIP and *p* value of different volatile compounds in different varieties of plum.

| Compound Type | Compound | F vs. C (23) | | C vs. K (121) | | F vs. K (108) | |
|---|---|---|---|---|---|---|---|
| | | VIP Value | *p* Value | VIP Value | *p* Value | VIP Value | *p* Value |
| Ester | Benzoic acid, methyl ester | 1.694 | 0.008 | 1.067 | 0.003 | 1.070 | 0.001 |
| | Butanoic acid, hexyl ester | 1.607 | 0.033 | - | - | - | - |
| | 2,6-Octadienoic acid, 3,7-dimethyl-, methyl ester | 1.649 | 0.034 | 1.070 | 0.006 | 1.073 | 0.019 |
| | Methyl acetate | 1.647 | 0.015 | 1.070 | 0.012 | 1.074 | 0.001 |
| | Ethyl acetate | 1.643 | 0.024 | 1.071 | 0.004 | 1.074 | 0.009 |
| | Bornyl acetate | - | - | 1.072 | 0.036 | 1.074 | 0.006 |
| | Citronellyl isobutyrate | - | - | 1.070 | 0.042 | 1.074 | 0.023 |
| | Linalyl acetate | - | - | 1.068 | 0.001 | 1.072 | 0.001 |
| | Diethyl Phthalate | - | - | 1.061 | 0.005 | - | - |
| | Benzeneacetic acid, 4-(1,1-dimethylethyl)-, methyl ester | - | - | - | - | 1.068 | 0.002 |

**Table 5.** *Cont.*

| Compound Type | Compound | F vs. C (23) | | C vs. K (121) | | F vs. K (108) | |
|---|---|---|---|---|---|---|---|
| | | VIP Value | *p* Value | VIP Value | *p* Value | VIP Value | *p* Value |
| | L-Aspartic acid, N-acetyl-, dimethyl ester | - | - | 1.058 | 0.015 | - | - |
| | 1,3-Cyclohexadiene-1-carboxylic acid, 2,6,6-trimethyl-, ethyl ester | - | - | 1.069 | 0.043 | - | - |
| | Carbamic acid, 2-chloroethyl ester | - | - | 1.050 | 0.016 | - | - |
| | Propanoic acid, 2-methyl-, 2-phenylethyl ester | - | - | 1.071 | 0.049 | 1.074 | 0.049 |
| | Benzyl tiglate | - | - | 1.071 | 0.011 | 1.074 | 0.010 |
| | cis-3-Hexenyl isovalerate | - | - | 1.071 | 0.039 | 1.074 | 0.039 |
| | Isobornyl formate | - | - | 1.070 | 0.036 | 1.074 | 0.009 |
| | Acetic acid, (propylthio)-, methyl ester | - | - | 1.066 | 0.001 | - | - |
| | 2-Butenoic acid, hexyl ester | - | - | 1.070 | 0.022 | 1.073 | 0.001 |
| | Acetic acid, phenyl ester | - | - | 1.070 | 0.009 | 1.073 | 0.026 |
| | 2-Oxepanone | - | - | 1.070 | 0.002 | 1.073 | 0.003 |
| | Ethyl 2-hexenoate, trans- | - | - | 1.071 | 0.037 | - | - |
| | cis-3-Hexenyl-.alpha.-methylbutyrate | - | - | 1.071 | 0.040 | 1.074 | 0.025 |
| | Methyl 6,6-dimethylbicyclo[3.1.1]hept-2-ene-2-carboxylate | - | - | - | - | 1.070 | 0.009 |
| | Benzoic acid, 1-methylethyl ester | - | - | - | - | 1.074 | 0.032 |
| | Benzoic acid, 2-propenyl ester | - | - | - | - | 1.074 | 0.010 |
| | Butanoic acid, 1,1-dimethyl-2-phenylethyl ester | - | - | - | - | 1.074 | 0.017 |
| Terpenoids | 2-Buten-1-one, 1-(2,6,6-trimethyl-1,3-cyclohexadien-1-yl)-, (E)- | 1.686 | 0.010 | - | - | - | - |
| | 2,6,10-Dodecatrien-1-ol, 3,7,11-trimethyl-, (Z,E)- | - | - | 1.059 | 0.023 | - | - |
| | Germacrene D | - | - | 1.071 | 0.020 | 1.074 | 0.020 |
| | cis-.alpha.-Bisabolene | - | - | 1.069 | 0.006 | 1.073 | 0.006 |
| | Naphthalene, decahydro-1,6-bis(methylene)-4-(1-methylethyl)-, (4.alpha.,4a.alpha.,8a.alpha.)- | - | - | 1.059 | 0.008 | - | - |
| | Humulene | - | - | 1.063 | 0.006 | 1.055 | 0.032 |
| | (S,1Z,6Z)-8-Isopropyl-1-methyl-5-methylenecyclodeca-1,6-diene | - | - | 1.061 | 0.026 | - | - |
| | (-)-trans-Isopiperitenol | - | - | 1.031 | 0.015 | 1.038 | 0.040 |
| | Bicyclo[2.2.1]heptane, 2-methyl-3-methylene-2-(4-methyl-3-pentenyl)-, (1S-endo)- | - | - | 1.063 | 0.007 | 1.054 | 0.030 |
| | (1S,4aR,7R)-1,4a-Dimethyl-7-(prop-1-en-2-yl)-1,2,3,4,4a,5,6,7-octahydronaphthalene | - | - | 1.064 | 0.004 | 1.056 | 0.030 |
| | Benzene, 1-(1,5-dimethyl-4-hexenyl)-4-methyl- | - | - | 1.071 | 0.007 | 1.074 | 0.007 |
| | Iridomyrmecin | - | - | 1.063 | 0.007 | 1.058 | 0.018 |
| | Isoborneol | - | - | 1.071 | 0.026 | 1.074 | 0.026 |
| | Citronellol | - | - | 1.063 | 0.003 | - | - |
| | 2,3-Dehydro-1,8-cineole | - | - | 1.071 | 0.005 | 1.074 | 0.005 |

**Table 5.** *Cont.*

| Compound Type | Compound | F vs. C (23) | | C vs. K (121) | | F vs. K (108) | |
|---|---|---|---|---|---|---|---|
| | | VIP Value | *p* Value | VIP Value | *p* Value | VIP Value | *p* Value |
| | .beta.-Phellandrene | - | - | 1.061 | 0.001 | 1.063 | 0.006 |
| | D-Carvone | - | - | 1.071 | 0.015 | 1.074 | 0.015 |
| | Carvone | - | - | 1.071 | 0.004 | 1.074 | 0.014 |
| | .gamma.-Terpinene | - | - | 1.071 | 0.003 | - | - |
| | Caryophyllene | - | - | 1.071 | 0.017 | - | - |
| | (2S,4R)-4-Methyl-2-(2-methylprop-1-en-1-yl)tetrahydro-2H-pyran | - | - | 1.071 | 0.032 | - | - |
| | Aristolochene | - | - | 1.071 | 0.047 | 1.074 | 0.022 |
| | Cyclohexene, 4-[(1E)-1,5-dimethyl-1,4-hexadien-1-yl]-1-methyl- | - | - | - | - | 1.067 | 0.047 |
| | 1H-Cyclopenta[1,3]cyclopropa[1,2]benzene, octahydro-7-methyl-3-methylene-4-(1-methylethyl)-, [3aS-(3a.alpha.,3b.beta.,4.beta.,7.alpha.,7aS*)]- | - | - | - | - | 1.066 | 0.012 |
| | Cyclohexanol, 5-methyl-2-(1-methylethyl)-, (1.alpha.,2.alpha.,5.alpha.)- | - | - | - | - | 1.048 | 0.008 |
| | 3,6-dihydro-4-methyl-2-(2-methyl-1-propenyl)- | - | - | - | - | 1.070 | 0.010 |
| | 1,3-Cyclohexadiene-1-carboxaldehyde, 2,6,6-trimethyl- | - | - | - | - | 1.073 | 0.045 |
| | 1,6-Octadiene, 3,7-dimethyl- | - | - | - | - | 1.075 | 0.011 |
| Hydrocarbons | Tridecane | 1.581 | 0.008 | 1.071 | 0.001 | 1.073 | 0.024 |
| | 1,4-Cyclohexadiene, 1-methyl- | 1.609 | 0.016 | 1.070 | 0.014 | 1.074 | 0.010 |
| | 1,5-Heptadiene, 2-methyl-, (Z)- | 1.534 | 0.017 | 1.070 | 0.023 | 1.074 | 0.009 |
| | Heptadecane, 2-methyl- | - | - | 1.071 | 0.042 | 1.074 | 0.042 |
| | Cyclohexane, 1-(1,5-dimethylhexyl)-4-methyl- | - | - | 1.063 | 0.002 | 1.058 | 0.032 |
| | 1-Pentadecene | - | - | 1.071 | 0.001 | 1.074 | 0.001 |
| | 1-Dodecene | - | - | 1.068 | 0.006 | - | - |
| | 4-Undecene, 3-methyl-, (Z)- | - | - | 1.069 | 0.017 | 1.073 | 0.002 |
| | Albene | - | - | 1.071 | 0.026 | 1.075 | 0.026 |
| | Decane, 4-methyl- | - | - | 1.069 | 0.026 | - | - |
| | Cyclohexane, (1,1-dimethylpropyl)- | - | - | 1.071 | 0.017 | 1.074 | 0.017 |
| | Bicyclo(3.3.1)non-2-ene | - | - | 1.068 | 0.004 | 1.072 | 0.002 |
| | Undecane, 2,9-dimethyl- | - | - | 1.071 | 0.005 | - | - |
| | Undecane, 2,4-dimethyl- | - | - | 1.070 | 0.023 | 1.073 | 0.043 |
| | Cyclohexene, 3,4-diethenyl-1,6-dimethyl- | - | - | 1.071 | 0.039 | 1.074 | 0.003 |
| | Octane, 2,3,3-trimethyl- | - | - | - | - | 1.073 | 0.038 |
| | (1S,5S)-2-Methyl-5-((R)-6-methylhept-5-en-2-yl)bicyclo[3.1.0]hex-2-ene | - | - | - | - | 1.074 | 0.035 |
| | 1-Undecene, 9-methyl- | - | - | - | - | 1.074 | 0.003 |
| Alcohol | 1,2-Benzenedimethanol | 1.679 | 0.007 | - | - | 1.064 | 0.001 |
| | 1,5,7-Octatrien-3-ol, 3,7-dimethyl- | 1.678 | 0.043 | - | - | - | - |
| | 5,9-Undecadien-2-ol, 6,10-dimethyl- | - | - | 1.071 | 0.016 | 1.074 | 0.016 |
| | 1-Cyclohexene-1-propanol, .alpha.,2,6,6-tetramethyl- | - | - | 1.063 | 0.003 | 1.066 | 0.005 |
| | 3-Buten-2-ol, 4-(2,6,6-trimethyl-2-cyclohexen-1-yl)-, (3E)- | - | - | 1.063 | 0.010 | - | - |

**Table 5.** *Cont.*

| Compound Type | Compound | F vs. C (23) | | C vs. K (121) | | F vs. K (108) | |
|---|---|---|---|---|---|---|---|
| | | VIP Value | *p* Value | VIP Value | *p* Value | VIP Value | *p* Value |
| | 1-Dodecanol | - | - | 1.051 | 0.042 | - | - |
| | Benzenemethanol, .alpha.-methyl- | - | - | 1.067 | 0.002 | - | - |
| | Cyclooctyl alcohol | - | - | 1.057 | 0.001 | - | - |
| | 3-Cyclopentyl-1-propanol | - | - | 1.070 | 0.041 | - | - |
| | 2-Pentyn-1-ol | - | - | 1.071 | 0.004 | - | - |
| | 1,2-Benzenedimethanol | - | - | - | - | 1.064 | 0.013 |
| | 1-Cyclopentene-1-methanol, 2-methyl-5-(1-methylethyl)- | - | - | - | - | 1.074 | 0.014 |
| | 2-Nonen-1-ol | - | - | - | - | 1.075 | 0.021 |
| | 7-Octene-2,6-diol, 2,6-dimethyl- | - | - | - | - | 1.075 | 0.011 |
| Ketone | 1-Butanone, 2-hydroxy-1-phenyl- | 1.693 | 0.007 | - | - | - | - |
| | 2-Propanone, 1-(4-methoxyphenyl)- | 1.688 | 0.010 | - | - | - | - |
| | 3-Nonen-5-one | 1.492 | 0.018 | 1.069 | 0.038 | 1.074 | 0.007 |
| | 3-Penten-2-one, 4-methyl- | 1.616 | 0.017 | 1.070 | 0.012 | 1.074 | 0.010 |
| | 6-Methyl-6-(5-methylfuran-2-yl)heptan-2-one | - | - | 1.067 | 0.030 | - | - |
| | 2-Butanone, 4-(2,6,6-trimethyl-1-cyclohexen-1-yl)- | - | - | 1.069 | 0.030 | 1.072 | 0.038 |
| | 2(1H)-Naphthalenone, 3,4,4a,5,6,7-hexahydro-1,1,4a-trimethyl- | - | - | 1.071 | 0.001 | 1.074 | 0.001 |
| | 3,4-Hexanedione, 2,2,5-trimethyl- | - | - | 1.063 | 0.029 | 1.068 | 0.002 |
| | Acetophenone | - | - | 1.068 | 0.002 | - | - |
| | 1(2H)-Naphthalenone, octahydro-4-hydroxy-, trans- | - | - | 1.064 | 0.003 | 1.055 | 0.040 |
| | 2-Cyclohexen-1-one, 4-hydroxy-3-methyl-6-(1-methylethyl)-, trans- | - | - | 1.071 | 0.041 | 1.074 | 0.041 |
| | 2H-Pyran-2-one, 6-pentyl- | - | - | 1.071 | 0.009 | 1.074 | 0.009 |
| | Cyclooctyl alcohol | - | - | 1.057 | 0.001 | - | - |
| | 3-Octen-2-one, (E)- | - | - | 1.051 | 0.045 | - | - |
| | Ethanone, 1-(2-hydroxy-4-methoxyphenyl)- | - | - | 1.071 | 0.047 | - | - |
| | spiro[3-oxatricyclo[3.3.1.02,4]nonane-8,1′-cyclopropane]-6-one | — | — | 1.069 | 0.018 | 1.074 | 0.043 |
| | 1,2-Cyclohexanedione | - | - | - | - | 1.073 | 0.031 |
| Aldehyde | 10-Undecenal | 1.508 | 0.014 | 1.070 | 0.028 | 1.072 | 0.037 |
| | 2-Hexenal, (Z)- | 1.630 | 0.016 | 1.070 | 0.011 | 1.074 | 0.010 |
| | 3-MethoxycinnamAldehyde | - | - | 1.059 | 0.008 | 1.058 | 0.019 |
| | 3-(4-Hydroxyphenyl)propanal | - | - | 1.070 | 0.005 | 1.074 | 0.005 |
| | 5-Heptenal, 2,6-dimethyl- | - | - | 1.070 | 0.026 | 1.074 | 0.022 |
| | BenzAldehyde, 3-methoxy- | - | - | 1.070 | 0.046 | - | - |
| | BenzAldehyde diethylacetal | - | - | 1.070 | 0.043 | - | - |
| | Tridecanal | - | - | - | - | 1.067 | 0.008 |
| | 2,6-Nonadienal, (E,Z)- | - | - | - | - | 1.074 | 0.019 |
| Acid | Acetic acid, phenoxy- | 1.686 | 0.004 | - | - | 1.068 | 0.003 |
| | 2-Hexenoic acid, (E)- | - | - | - | - | 1.074 | 0.001 |
| Phenol | Methyleugenol | 1.649 | 0.038 | 1.070 | 0.016 | 1.074 | 0.028 |
| | Phenol, 2-methyl-5-(1-methylethyl)- | - | - | 1.071 | 0.023 | 1.074 | 0.023 |
| | Phenol, 2-methyl- | - | - | 1.071 | 0.004 | 1.074 | 0.043 |
| | Phenol, 2,3,6-trimethyl- | - | - | - | - | 1.074 | 0.011 |
| Amine | Butyramide, 2-cyano-2-ethyl- | - | - | 1.071 | 0.019 | - | - |
| | Hordenine | - | - | 1.071 | 0.048 | 1.074 | 0.019 |

| Compound Type | Compound | F vs. C (23) | | C vs. K (121) | | F vs. K (108) | |
|---|---|---|---|---|---|---|---|
| | | VIP Value | *p* Value | VIP Value | *p* Value | VIP Value | *p* Value |
| Aromatics | Benzene, 1-methyl-4-(1-methylethenyl)- | 1.673 | 0.023 | - | - | - | - |
| | Benzene, 1,2-dimethoxy-4-(1-propenyl)- | - | - | 1.071 | 0.025 | 1.075 | 0.025 |
| | 2-Methoxy-4-vinylphenol | - | - | 1.071 | 0.034 | 1.075 | 0.034 |
| | Benzoic acid, 4-hydroxy- | - | - | 1.071 | 0.006 | 1.074 | 0.006 |
| | Indan, 1-methyl- | - | - | 1.071 | 0.027 | 1.074 | 0.027 |
| | Benzene, 1-methyl-2-(1-ethylpropyl)- | - | - | 1.071 | 0.046 | 1.074 | 0.002 |
| Heterocyclic | Furan, 2-butyltetrahydro- | 1.601 | 0.004 | 1.069 | 0.015 | 1.074 | 0.004 |
| | 2,2′-Ethylidenebis(5-methylfuran) | 1.688 | 0.010 | - | - | - | - |
| | Thiophene, 3-methyl- | 1.638 | 0.016 | 1.070 | 0.010 | 1.074 | 0.010 |
| | 8-Azabicyclo[3.2.1]octan-3-ol, 8-methyl-, endo- | - | - | 1.065 | 0.012 | - | - |
| | Meperidine | - | - | 1.061 | 0.002 | 1.069 | 0.005 |
| | 2-Methyl-1,3-dithiacyclopentane | - | - | 1.063 | 0.026 | - | - |
| | Ethanone, 1-(2-pyridinyl)- | - | - | 1.069 | 0.003 | - | - |
| | 1,2,4,5-Tetrazin-3-Amine | - | - | 1.068 | 0.001 | - | - |
| | 2H-Pyran-2-one, tetrahydro- | - | - | 1.068 | 0.023 | - | - |
| | 2-PyridinemethanAmine | - | - | 1.070 | 0.020 | 1.074 | 0.035 |
| | 2(5H)-Furanone, 5-ethyl-3-hydroxy-4-methyl- | - | - | 1.069 | 0.007 | 1.074 | 0.010 |
| | 4H-Pyran-4-one, 5-hydroxy-2-(hydroxymethyl)- | - | - | 1.063 | 0.002 | 1.056 | 0.044 |
| | 2-Ethoxy-3-methylpyrazine | - | - | 1.069 | 0.008 | 1.073 | 0.005 |
| | 5H-5-Methyl-6,7-dihydrocyclopentapyrazine | - | - | 1.071 | 0.028 | 1.074 | 0.028 |
| | 1H-Pyrazole, 1,3,5-trimethyl- | - | - | 1.068 | 0.001 | - | - |
| | 1H-1,2,4-Triazole, 3-chloro- | - | - | 1.071 | 0.023 | 1.075 | 0.018 |
| | Ethanone, 1-(1H-pyrrol-2-yl)- | - | - | 1.072 | 0.015 | 1.074 | 0.015 |
| | 3-Acetyl-2-oxo-1,3-oxazolidine | - | - | 1.070 | 0.029 | - | - |
| | Pyrazine, tetramethyl- | - | - | 1.071 | 0.016 | - | - |
| | 4-Piperidinone, 1,3-dimethyl- | - | - | 1.071 | 0.030 | 1.073 | 0.018 |
| | 4-Methylthiazole | - | - | 1.071 | 0.029 | - | - |
| | 2-((3,3-Dimethyloxiran-2-yl)methyl)-3-methylfuran | - | - | 1.071 | 0.037 | 1.074 | 0.004 |
| | 2-(3-Thienyl)pyridine | - | - | - | - | 1.074 | 0.000 |
| | Furaneol | - | - | - | - | 1.073 | 0.007 |
| Nitrogen compounds | 2,6-Octadienenitrile, 3,7-dimethyl-, (Z)- | - | - | 1.071 | 0.049 | 1.074 | 0.049 |
| | 2,6-Octadienenitrile, 3,7-dimethyl-, (Z)- | - | - | 1.070 | 0.035 | 1.074 | 0.007 |
| | Cyclopentanecarbonitrile, 3-(1-methylethylidene)- | - | - | 1.071 | 0.040 | 1.074 | 0.048 |
| Sulfur compounds | Diallyl disulphide | - | - | 1.071 | 0.015 | - | - |

## 4. Conclusions

The volatile compounds of three varieties of plum fruits, 'Fengtang' plum, 'Kongxin' plum, and 'Cuihong' plum, were qualitatively and quantitatively analyzed by HS-SPME-GC-MS. A total of 938 volatile components were detected and identified, with a total of 470 volatile substances. It was found that the volatile substances in the fruits of the three varieties of plum mainly consisted of 14 classes of substances, including terpenoids,

esters, heterocyclics, hydrocarbons, ketones, alcohols, aldehydes, aromatic hydrocarbons, amines, acids, phenols, nitrogenous compounds, sulfur-containing compounds, and other categories. And it was shown that benzene, 1,2-dimethoxy-4-(1-propenyl)-, benzoic acid, 4-hydroxy-, [1$\alpha$,4a$\alpha$,8a$\alpha$]-1,2,4a,5,6,8a-hexahydro-4-7-dimethyl-1-[1-methylethyl]naphthalene, Isoborneol, 2,6-Nonadienal, (E,Z)-, and Benzyl tiglate could be used to differentiate the main substances of the 'Kongxin' plum. In addition, F vs. C, C vs. K, and F vs. K screened out 6, 46, and 40 substances with highly significant differences, respectively, and were dominated by terpenoids and esters, which can be used as marker volatile compounds for distinguishing the fruits of the three different varieties of plum. These findings provide an important scientific basis for the scientific evaluation of the volatile quality of plum fruits, as well as the tracing of the origin and variety of plum fruits through the volatile components.

**Author Contributions:** X.D. and Y.M. conceived the project; Q.Z. and S.Z. designed and performed the experiments; Q.Z. wrote the paper; X.L. was responsible for data curation, formal analysis, and methodology; X.D. was responsible for supervision and writing—review and editing; Y.Z. and X.W. were responsible for data curation and validation; J.P. and D.L. were responsible for visualization and methodology. All authors discussed the results and commented on the manuscript. All authors have read and agreed to the published version of the manuscript.

**Funding:** This research was funded by the National Natural Science Foundation of China, grant number 31960620 and the Guizhou Provincial Science and Technology Plan of Key Supporting Project (2022) 018.

**Data Availability Statement:** All data are included in the manuscript.

**Conflicts of Interest:** The authors declare no conflict of interest.

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
