# Peer review of "Analysis of Volatile Compounds in Different Varieties of Plum Fruits Based on Headspace Solid-Phase Microextraction-Gas Chromatography-Mass Spectrometry Technique"

_horticulturae, doi:10.3390/horticulturae9101069_

Round 1
Reviewer 1 Report
It is a good work in which they propose to validate the HS-SPME-GC-MS technique to analyze volatile compounds of plum fruits and to be able to use it in the characterization and identification of different varieties and their relationship with flavor, quality and geographical origin. However, I suggest that they carry out a sensory evaluation with an untrained panel that allows validating those differences found instrumentally, with possible sensory differences perceived by consumers.
In addition, I present the following specific observations that should be addressed to improve the understanding of the manuscript:
L41, L45: After full stop, it must start with a capital letter. Correct the word 'Volatile'.
L52-55: Studies by Li et al. they were made with Cerasus humilis (Bge.) sok and not with Prunus serrulata. Please correct.
L59.62: Reference [14], is not from Wei. Please correct the paragraph or put Wei's reference in the references section.
L65-77: In this paragraph the information is repetitive, please improve the wording.
L102-103: "containing saturated sodium chloride solution" should describe the amount of saturated sodium chloride solution used.
L131: The probability symbol must be in lowercase (p<0.05 or p<0-01). Please correct throughout the document (Table 5, too).
L140-151: It is tiring to read exactly the same results that are in Table 1, having them relatively close.I suggest that you describe the most relevant results in Table 1., For example: There were no differences in the amount and type of volatile compounds between the three varieties (this is indicated by the absence of literals by treatment), etc.
Table 1. You must describe the meaning of the subtitles 'Total' and 'Total' that cause confusion or doubt. For example, the first should read 'Total volatile compounds' and the second 'Total volatile common substances'.
L168-170: The information presented as 'Note' in Table 1 is irrelevant, since it is not showing average values and S.E., nor does it show literals or * that denote differences between treatments and genotypes? in any case they would be varieties. Please remove the note.
L174-175: According to the data presented in Table 2, the sum of these three types of compounds are between 65.8 and 68.7% among the three varieties. Please check and correct the wording.
L178: "were mainly produced by the carbohydrate metabolic pathway [15]" Having not done carbohydrate metabolic studies, he cannot be certain that carbohydrates were the precursors of terpenoids; furthermore, upon reviewing the citation manuscript [15], they also did not study carbohydrate metabolism and do not make any inferences that the two terpenes found in teas were generated by carbohydrate metabolism. Please correct the wording.
L183: " with the content of ‘Cuihong’ plum > ‘Fengtang’ plum> ‘Kongxin’ plum." It is an incorrect estimate. According to the results obtained it should be: "‘Cuihong’ plum = ‘Fengtang’ plum> ‘Kongxin’ plum." Since there were no statistically significant differences between Cuihong and Fengtang plum. Therefore, the numerical values cannot be used to want to denote a better effect in a variety. Please review the document (L186) and correct according to the results obtained.
L185: "were mainly derived from the metabolism of amino acids [17]" Similar case, this process cannot be fully confirmed, since they did not carry out metabolic analyzes of amino acids. They can assume that this could be a possible pathway for ester formation, but not state it categorically. Please correct the wording. The same correction must be made with the other compounds (L186-188, etc.), since it ensures routes and pathways of formation that are not demonstrated in this investigation.
L197-201: Repetitive paragraph. Please delete it or change the wording to a better discussion of the most outstanding results.
Table 2: Here if applicable, put in the Table Footnote: "Data are showed as the mean ± S.E." please include it.
L235: It should explain what aromatic connotation Carvenone brings, allowing for a proper discussion on aroma or flavor (as it does on the percentage difference) between 'Fengtang' plum vs 'Cuihong' plum. Or why or why do these Carvenone differences stand out between these two varieties? The same should apply to the other substances mentioned in the same paragraph.
Table 3: In the values that the compound was not detected, instead of using '-', you should put LOD: limit of detection. When it was not detected. And describe the abbreviation in Table Note, as well as the meaning of the other abbreviations used in the Table (CAS, RT).
Figure 4: Very bad resolution, the texts cannot be read. please improve it.
L299: The results in Table 5 show different values, please check and correct the text or the Table.
L331: "Conclsusions" Spelling error in the word, please correct it by 'Conclusions'
CONCLUSIONS: The wording that appears in this section are not conclusions of the work, they are a repetition of the results already described in the corresponding section. Please remove them from this section and focus on writing the conclusion in accordance with the title and objective set forth in the document. To my knowledge, the information from L351 ("terpenes and esters, which can be used as important markers of differential volatile substances for distinguishing the fruits of the three different varieties of plum. This study can provide an important scientific basis for the scientific evaluation of the volatile quality of plum fruits, as well as the tracing of the origin and variety of plum fruits through the volatile compounds."), could be the basis for formulating the conclusions of this study responding to the objective that was set . Please improve the conclusions.
REFERENCES: reference [16] is not mentioned in the body of the document. Please remove it from this section.
Author Response
请参阅附件。

Reviewer 2 Report
Dear Authors,
I have examined your paper and the article is hard to follow, the language needs improvements to be readable.
The subject of the study is in the scope of the journal.
- The resolution of the figures used in the manuscript is very low. All figures must be rearranged.
- There is no information about "reagents" in the materials and methods section. It must be added.
Other revisions are stated on the text.
Regards,

The article is hard to follow, the language needs improvements to be readable.
Reviewer 3 Report
Manuscript recieved for review analyze different varieties of plum for volatile substances by head-space solid-phase microextraction combined with gas chromatography-mass spectrometry.
Title and abstract are adequate.
Intorduction section is elatorate enough, with explanation of researched topic. The aim of the study is detailed.
Material and methodes section is appropriate for the described and conducted testing.
Results and discussion section is adequate and sufficient. Quality of presented results is very good, with very elaborate discussion and referencting to other authors’ results. Some minor corrections of the presented figures are noted in manuscripts’ pdf file.
Conclusion section is appropriate for the presented results.
Decission: minor revision

Author Response
请参阅附件。

Round 2
Reviewer 1 Report
The manuscript entitled "Analysis of volatile compounds in different varieties of plum fruits based on HS-SPME-GC-MS technique", was considerably improved with the corrections made, so it can be considered for publication in this Journal. However, you should give it a general spelling check as I still detected some minor errors and details that should be taken into account.
L85: "placed at 4 ℃and 85%-90%" Correct spelling error: separate the unit of measurement from the word 'and'
Table 2: Please homogenize the number of decimals in the Hydrocarbons value of 'Kongxin' plum: "5.54" by '5.540'
Table 3: In the footnote of the table you must add: "Data are shown as the mean±S.E. Different letters in the same row differ significantly (p<0.05)"
L295-296: "23, 121 and 108 differential volatile substances were screened in F vs C, C vs K and F vs K, respectively" The same error that I had already pointed out continues. In Table 5, the values shown are: "F vs C (23) C vs K (108) F vs K (97)" Which does not agree with the values that are in the wording. Please verify and correct.
Author Response
请参阅附件。
